# Improving the Analysis of Sulfur Content and Calorific Values of Blended Coals with Data Processing Methods in Laser-Induced Breakdown Spectroscopy

**Jae Seung Choi [1], Choong Mo Ryu [1], Jung Hyun Choi [2] and Seung Jae Moon [1,\*]**

[1] Department of Mechanical Convergence Engineering, Hanyang University, Seoul 04763, Republic of Korea
[2] Department of Environmental Science and Engineering, Ewha Womans University, Seoul 03760, Republic of Korea
\* Correspondence: smoon@hanyang.ac.kr

**Featured Application: In Situ component and calorific value analysis of mixed coals in a thermal power plant.**

**Abstract:** In Situ monitoring of the calorific value of coal has the advantage of reducing the amount of unburned carbon by injecting an appropriate amount of combustion air immediately to induce complete combustion. High sulfur concentrations cause severe environmental problems such as acid rain. In order to estimate the calorific value and measure the sulfur concentration, a new powerful technique for mixed coals was studied. Laser-induced breakdown spectroscopy (LIBS) does not require sample preparation. Several blended coals were used for the experiment to replicate the actual coal-fired power plant conditions. Two well-known data processing methods in near-infrared spectroscopy have been adopted to enhance the weak sulfur emission lines. The performance of the partial least square regression model was established by the parameters such as coefficient of determination, $R^2$, relative error, and root mean square error (RMSE). The RMSE average was compared with the results of previous studies. As a result, the values from this study were smaller by 6.02% for the calibration line and by 4.5% for the validation line in near-infrared spectroscopy. The RMSE average values for calorific values were calculated to be less than 1%.

**Keywords:** laser-induced breakdown spectroscopy (LIBS); coal; sulfur; calorific value; multivariate data processing

## 1. Introduction

The environmental impact of coal-fired power generation is growing. Korea is still considerably dependent on coal as an energy source due to price competition. The contribution ratio of coal-fired power plants to total emission by air pollutants is 2.7% for total suspended particles, 3.7% for particle matter 10 (PM10), 4.7% for particle matter 2.5 (PM2.5), 16.2% for nitrogen oxides (NOx), and 24.1% for sulfur oxides (Sox) in Korea, according to the Korea national institute of environmental research data [1]. According to this report, the summation of SOx and NOx has the largest proportion, at over 40%. In the case of NOx emitted from coal-fired power plants, most of the combined nitrogen comes from the combustion air and should be controlled through a denitrification facility. On the other hand, in the case of Sox, the use of fuel containing a small amount of sulfur is recommended, because the sulfur comes from the fuel itself [2]. By using in situ measurement techniques for determining the calorific value of the coal and sulfur concentration within the coal, it can be possible to control of the excessive amounts of

combustion air and warnings can be given about excessive sulfur concentrations in the non-uniformly mixed coal.

Conventional techniques for quantitatively analyzing element concentrations usually need complicated sample pretreatment such as fusion dissolution and microwave digestion and require standard reference materials. Due to these problems, analyzing many samples can be expensive and quite time-consuming [3]. Thus, a fast and reliable analysis method is required. To overcome these limitations of conventional analysis methods, many studies have been conducted measuring the element concentration of coal using spectroscopy methods. Bona et al. [4] conducted a study to measure the values of coal using mid-infrared spectroscopy. In this study, the results of three modes of the multiplicative scatter correction (MSC) data preprocessing method were detailed. The minimum value of the average root mean square error (RMSE) of sulfur was 37.36%, which was somewhat high. Wang et al. [5] conducted a near-infrared (NIR) spectroscopy study on the measurement of various properties from four types of coal by applying multivariate statistical techniques. The minimum value of the percentage error of the mean RMSE of sulfur was 8.98% for full sample sets, and the averaged mean RMSE of all kinds of the coal samples was 13.53%. These values were considerably lower than those of the mid-infrared spectroscopy study. In recent times, research on a technology called laser-induced breakdown spectroscopy (LIBS) have been actively conducted. LIBS is a spectroscopic method using lasers. Qualitative and quantitative analysis of elements is possible, including lighter elements in the periodic table, regardless of their states of gas, liquid or solid. A small quantity of mass on the sample surface is ablated by a focused high energy laser beam. The plasma is then generated and expanded above the sample surface. Discrete atomic lines are emitted after a cooling process [6]. This emitted light can be collected by a spectrometer and used for qualitative and quantitative elemental analysis. LIBS has been successfully applied in various fields, especially for element analysis. Martin et al. [7] quantitatively identified the elemental composition of preservative-treated wood using principal component analysis (PCA). In another study, Yao et al. [8] studied a set of fertilizer samples to reveal the correlation of phosphorus and potassium using the partial least square (PLS) regression analysis method. These studies suggest that elemental analysis with LIBS is feasible and can be applied to sulfur concentration analysis. In fact, some studies have attempted to measure the sulfur concentration using LIBS. Gaft et al. [9] tried to use single-pulse and double-pulse laser irradiation approaches to apply LIBS to on-line sulfur analysis of minerals under ambient conditions. However, data processing methods were not used to detect and improve the sulfur emission lines in their study.

From recent studies, it is possible to directly measure the composition of coal using LIBS as well as elemental analysis. Yao et al. [10] attempted to employ multivariate analysis to extract coal ash content information from LIBS spectra rather than from the concentrations of the main ash-forming elements. Gaft et al. [11] have made efforts to measure ash in real time by using LIBS. Dong et al. [12] tested the analytical methods with partial correlation and principal component regression to extract the correlation between the amount of volatile matter and the LIBS spectral information based on coal structure. Yuan et al. [13] applied the multivariate dominant factor based on the PLS model to demonstrate an overall improvement in performance compared with the conventional PLS model for various coal properties such as ash content, volatile matter content, and calorific value.

In spite of these previous research studies on coal calorific values, there still remains different perspective issues to be resolved. As more coal-fired power plants in Korea use up to four types of coal blending to cut costs; quantitative analysis of blended coal calorific values is needed. If only one type of coal is used in the coal-fired power plant, pre-analyzed data values can be used. However, using blended coal makes it difficult to analyze in real-time. Moreover, if the coal samples are not uniformly mixed in the conveyor system, heterogeneous distribution of sulfur can occur. The prediction of sulfur

concentration by LIBS in conjunction with PLSR, especially for blended coal samples, is the main concern of this work.

From other previous research, two challenges which can be major obstacles in detecting sulfur emission lines with LIBS have been identified [14,15]. The most important challenge is that the spectrometer of the LIBS system used in this study can only detect from 200 to 800 nm spectral range. The strongest intensity of sulfur emission lines usually can be detected beyond this limit, at 125–180 nm (vacuum ultraviolet range) [16] and longer than 900 nm (near-infrared range) [17]. When using LIBS, the 125–180 nm region cannot be measured, and the measurement accuracy is low in the region above 900 nm [9]. To overcome this problem, the data quality of weak sulfur ionic emission lines in the 400–600 nm region needs to be enhanced. Therefore, to determine the proper data processing method, two kinds of methods were investigated. These two methods have been employed in NIR spectroscopy studies but rarely in LIBS studies.

In this study, a reduction in the prediction error for sulfur analysis will be attempted by applying an appropriate data processing method with LIBS. The calorific values will be analyzed without using Dulong's equation. Ten original coal samples were blended with varying blending ratio. Therefore, sixty blended coal samples were used to construct the regression model in this work. Furthermore, one kind of coal sample that was not applied to consist of the calibration line was verified with our method utilized to estimate the sulfur concentration and calorific value of unknown samples. The blended samples mixed with the unknown original sample were investigated for verification.

## 2. Materials and Methods

### 2.1. Materials and Measuring Systems

Ten bituminous coals, produced in various mining sites in the world, and combusted in a coal-fired power plant in Korea, were Gunvor, Peabody, Whiteheaven, Noble, MacQuarie, Lanna Harita, Glencore, Carbo One, and two types of MSJ. The blended coal samples were prepared by mixing the above 10 kinds of coal. All coal samples were pulverized into a powder with a size less than 100 µm and pelletized with a varied mixing ratio. Lal et al. [18] proved that the pellet samples could provide the highest possible precision. In the case of the powder samples, the shock wave caused by the high energy laser pulse can interrupt the sample surface; consequently, the laser pulse was absorbed above the sample surface due to the flying debris. In this study, 0.3 g of each blended powder sample was pressed by approximately 10 tons for 2 min using a 13 mm diameter pelletizer machine. The sulfur concentrations and calorific values to be used in the partial least square regression (PLSR) analysis are listed in Table 1, with 60 samples from C1 to C60.

Figure 1 presents the analysis system for pelletized and blended coal samples using a J200-EC LIBS system (Applied Spectra Inc., Fremont, CA, USA). Axiom software controlled the LIBS system. The 4th harmonic Nd:YAG laser (1064 nm) irradiation with energy varying from 9.9 to 87.3 mJ was focused to a 100 µm-sized spot. All emission from the laser-induced plasma was collected using an optical fiber bundle with a 5-channel charge coupled device (CCD) spectrometer covering wavelengths from 190 to 890 nm. This instrument was equipped with a high efficiency particulate air (HEPA) filter that could purge the particles from the laser ablation chamber as well as an *xyz*-translational stage. In order to obtain the best signal/background ratio in this study, gate delay time and repetition rate were optimized at 1.4 µs and 1 Hz, respectively. The gate delay time is the difference between the laser pulse and the emission line collection of the spectrometer. For example, zero means that the spectrometer collects the data as soon as a laser pulse is initiated. At the initial stage, only small intensities were generated since a continuous spectrum was predominantly emitted. After a few microseconds, the peak emission lines became apparent. However, if the time delay is too long, the plasma cools down and the peak emission lines will not be distinguished. Therefore, it was important to set an

appropriate delay time [19]. Even though the coal samples were pulverized and blended, sample heterogeneity exists. In order to reduce this problem and shot-to-shot laser fluctuation, each blended coal sample pellet was ablated at forty-nine different locations using a (7 × 7) grid pattern on the sample surface with a laser pulse energy of 30 mJ. Averaged values of the data obtained from these forty-nine locations were used to determine the PLSR model.

To evaluate the performance of sulfur concentration and calorific value, blended coal samples were analyzed using the PLSR approach by the Unscrambler X-version 10.3 (CAMO) software program. In this program, previous multiple $Y$ responses were chosen to develop a PLSR model called the PLS2 method. In this work, sixty different blended coal samples with various sulfur concentrations from 0.46 to 1.44% were used as the calibration data set. The calorific value range of these samples was between 6360 and 7275 kcal/kg. This calibration model will be used as a good standard for unknown sample prediction. To evaluate the prediction ability and reproducibility, the remaining unknown original coal sample, which was not included in application of the PLSR model, was regarded as an unknown sample and used for regression of prediction data set.

**Table 1.** Reference concentrations of blended coal samples.

| Reference Concentration | | | | | | | | |
|---|---|---|---|---|---|---|---|---|
| Sample | S (%) | Calorific Value (kcal/kg) | Sample | S (%) | Calorific Value (kcal/kg) | Sample | S (%) | Calorific Value (kcal/kg) |
| C 1 | 1.3983 | 6567 | C 21 | 0.5600 | 6743 | C 41 | 0.5546 | 6861 |
| C 2 | 1.2866 | 6624 | C 22 | 0.5900 | 6866 | C 42 | 0.6291 | 6872 |
| C 3 | 1.1749 | 6681 | C 23 | 0.6200 | 6990 | C 43 | 0.7037 | 6882 |
| C 4 | 1.0632 | 6738 | C 24 | 0.6500 | 7113 | C 44 | 0.7782 | 6893 |
| C 5 | 0.9515 | 6795 | C 25 | 0.6800 | 7129 | C 45 | 0.8528 | 6904 |
| C 6 | 0.5183 | 6935 | C 26 | 0.6300 | 6580 | C 46 | 1.3383 | 6567 |
| C 7 | 0.5567 | 7020 | C 27 | 0.5900 | 6530 | C 47 | 1.1667 | 6623 |
| C 8 | 0.5950 | 7105 | C 28 | 0.5500 | 6480 | C 48 | 0.9950 | 6680 |
| C 9 | 0.6333 | 7190 | C 29 | 0.5000 | 6430 | C 49 | 0.8233 | 6737 |
| C 10 | 0.6717 | 7275 | C 30 | 0.4600 | 6380 | C 50 | 0.6517 | 6793 |
| C 11 | 0.5600 | 6621 | C 31 | 1.3284 | 6480 | C 51 | 0.7900 | 6500 |
| C 12 | 0.5800 | 6623 | C 32 | 1.1469 | 6450 | C 52 | 0.8000 | 6571 |
| C 13 | 0.6000 | 6625 | C 33 | 0.9653 | 6420 | C 53 | 0.8100 | 6641 |
| C 14 | 0.6300 | 6627 | C 34 | 0.7837 | 6390 | C 54 | 0.8200 | 6711 |
| C 15 | 0.6500 | 6629 | C 35 | 0.6022 | 6360 | C 55 | 0.8300 | 6782 |
| C 16 | 0.8500 | 6863 | C 36 | 1.4129 | 6577 | C 56 | 1.4413 | 6543 |
| C 17 | 0.8700 | 6873 | C 37 | 1.3158 | 6645 | C 57 | 1.3726 | 6575 |
| C 18 | 0.8800 | 6883 | C 38 | 1.2187 | 6712 | C 58 | 1.3039 | 6608 |
| C 19 | 0.9000 | 6894 | C 39 | 1.1215 | 6780 | C 59 | 1.2352 | 6640 |
| C 20 | 0.9100 | 6904 | C 40 | 1.0244 | 6847 | C 60 | 1.1665 | 6673 |

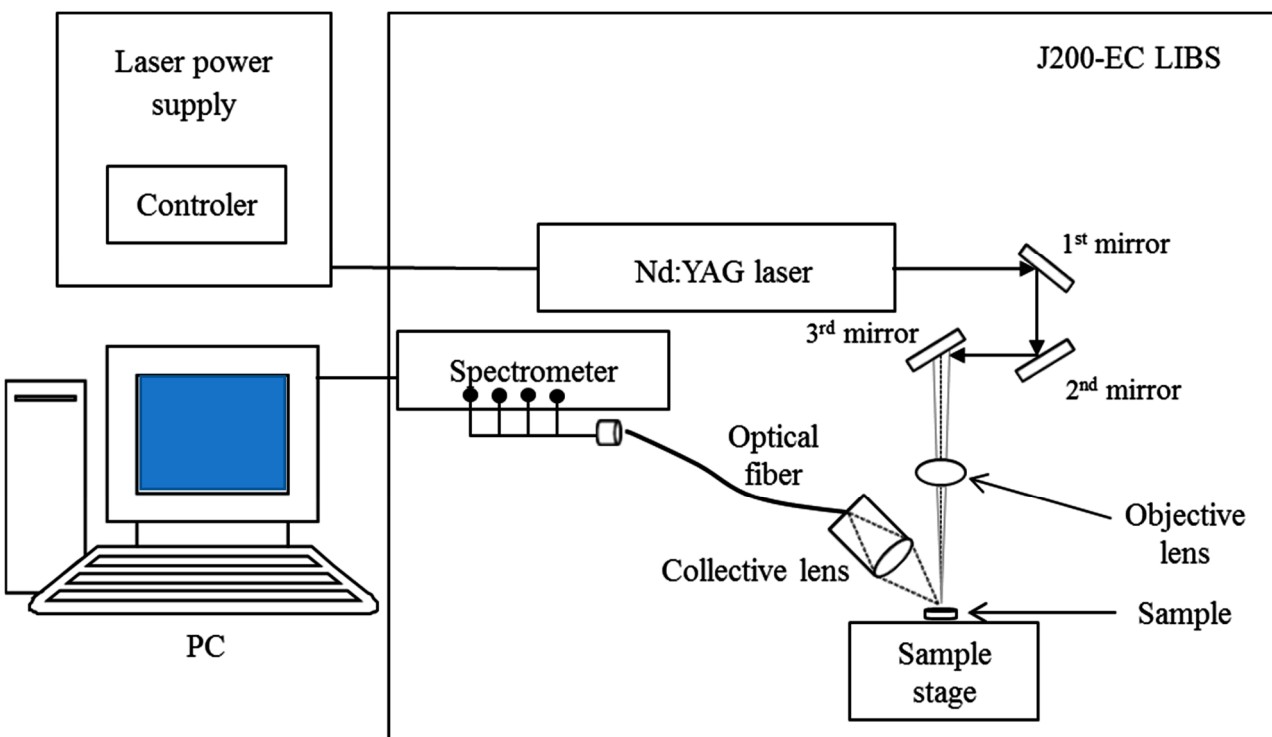

**Figure 1.** Schematic diagram of LIBS.

### 2.2. Statistical Analysis

In order to use the full range of the informative data and to eliminate noise, using a data processing method is essential for a robust calibration model. While the LIBS has been used for normalizing data treated with processing method in most cases, other spectroscopy methods such as NIR spectroscopy have been employed for several data processing methods to predict qualitative and quantitative content analysis.

In this study, Savitzkye-Golay (SG) smoothing and the multiplicative scatter correction (MSC) methods were used for the analysis of the concentration in the blended coal samples. The SG smoothing method uses linear least squares and fits sub-sets of adjacent data with a certain order of polynomial. The SG smoothing method can eliminate spectral noise effectively. It is important to properly adjust polynomial order and the number of smoothing points when using this method. Among this, the number of smoothing points in determining the degree of smoothing is very significant. If the number of smoothing points is too small, a calculation error will occur, resulting in poor model accuracy. Too many smoothing points will cause the spectral data to become too flat and less accurate. Therefore, it is important to test and decide what mode will produce the best results by changing the number of points [20]. Seven points and a third-order polynomial in the SG smoothing method were selected as a proper mode to avoid either calculation error or excessive smoothing. The MSC is another effective data processing method. This is used for the correction of non-uniform particle sizes, gap between particles, and uneven flatness of the sample surface. This method can modify the spectrum of each sample so that all samples have the same scattering signal regarded as an ideal spectrum [20].

### 2.2.1. Relative Standard Deviation (RSD)

Figure 2a shows the entire LIBS spectrum of mixed coal samples 4 and 34 as shown in Table 1. The peaks for carbon (247.9 nm), sulfur (416.3 nm), hydrogen (656.3 nm), and oxygen (777.4 nm) are clearly distinguished because they are abundantly present in coal samples. However, as shown in Figure 2b, the peak (416.267 nm) for sulfur is indistinguishable due to its small presence in the coal samples. The composition of S is

1.0632 and 0.7837 wt% in samples 4 and 34, respectively. The multivariable analysis can be performed based on the maximal intensity of each element.

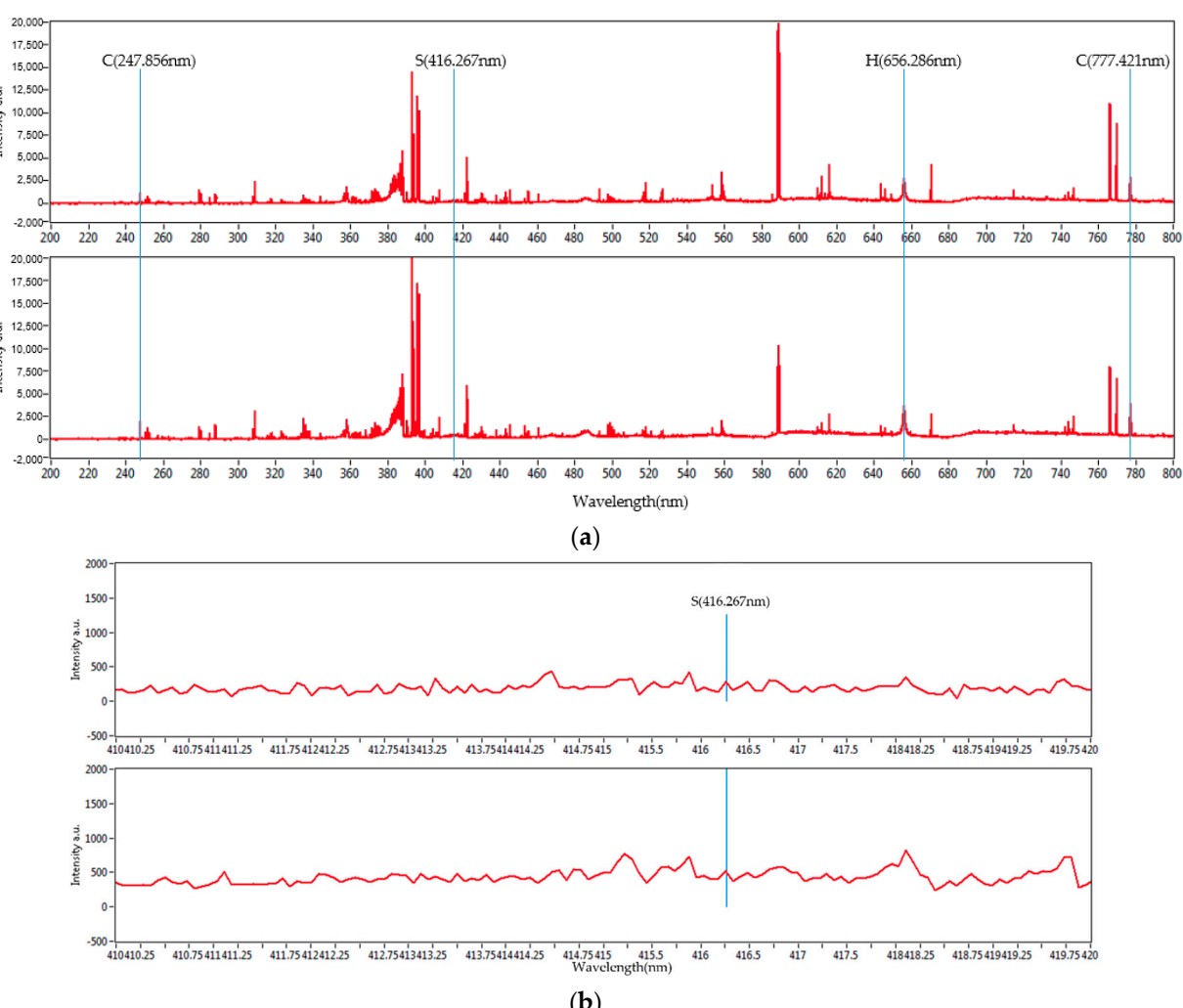

**Figure 2.** (**a**) LIBS spectrum of mixed coal samples 4 and 34, (**b**) sulfur peak of mixed coal samples 4 and 34.

It is important to reduce the peak noise of the sulfur emission line to construct a robust PLSR line. When the reproducibility is assessed, the relative standard deviation (RSD) is generally used as an appropriate measure. The RSD can be calculated by using the following equation [21]:

$$RSD = 100\% \times \left[ \sum (x_i - M)^2 / (n-1) \right]^{1/2} / M, \tag{1}$$

where *n*, $x_i$, and *M* are the number of a set for the measurement, the result of each measurement, and the arithmetic mean value of the set of repeated measurements, respectively. As the RSD is closer to zero, this means that the reproducibility is better.

### 2.2.2. Partial Least Square Regression (PLSR)

The PLSR model was adopted by using a full cross-validation method on the average recorded spectra [22]. The full spectrum range was employed in the model. In this study, the coefficient of determination, $R^2$, root mean square error of calibration (RMSEC), and root mean square error of cross-validation (RMSECV) were employed as the testing parameters for investigating the performance of PLSR. The RMSEC and RMSECV can be calculated by the following equation [10]:

$$RMSEC(V) = \sqrt{\frac{\sum_{i=1}^{n}(x_i - \hat{x}_i)}{n}} \tag{2}$$

where $n$, $\hat{x}_i$, and $x_i$ are the number of samples for calibration and validation, the reference concentration of the $i$th sample, and the predicted concentration of the $i$th sample, respectively. The RMSEC and RMSECV obtained from PLSR were used to accurately predict the sulfur concentration and calorific value of coal from the unknown samples. If these values are zero, they match the measured values. The closer the RMSE values are to zero, the better the model.

As a kind of multivariate analysis method, PLSR can provide a relationship between a set of predictor variables, $X$, and a set of response variables, $Y$. When the LIBS data is processed with PLSR, the predictor variables are the LIBS spectra intensities, and the sulfur concentrations are the response variables. The PLSR line is obtained in order to minimize the sum of the squared values of the differences between the measured value and the function value. Based on the PLSR model created in this part, the validity of the unknown sample prediction will be verified. As mentioned above, too many smoothing points and too high a polynomial order can cause loss of information. Since the even number-ordered polynomial was not different from the original due to the symmetric shape, an odd number-ordered polynomial was used. In order to obtain optimal results, results were compared by increasing the polynomial order and smoothing points. Therefore, in addition to the third-order polynomial, the fifth-order polynomial was tested by increasing the number of smoothing points to determine the proper SG smoothing mode that produces the best performance of the PLSR.

2.2.3. Root Mean Square Error (RSME) Average

To compare the RMSE quantitatively, the RMSE average concept was employed by the following equation [23]:

$$RMSE(avg.)(\%) = \frac{RMSE}{Average\ of\ Property} \times 100. \tag{3}$$

The RMSE average is calculated by dividing the RMSE by the average of reference values. The average of property means the average value of measured reference data that was used for the $Y$ value in the PLSR model.

**3. Results and Discussion**

*3.1. Sulfur Analysis*

3.1.1. Relative Standard Deviation (RSD)

Figure 3 represents the RSD results of the original and processed data by the two different methods for the major wavelength of sulfur, by using a bar chart. The major peak wavelength of sulfur (416.3 nm) line was selected for data analysis. This is because the signal to background ratio was the lowest at 416.3 nm. Fifteen out of the sixty samples showed the smallest RSD value when applying the MSC data processing method. The original data showed the smallest RSD value in only two samples. However, forty-three out of the sixty samples showed the lowest RSD value when the SG smoothing method was applied. Some samples showed a too large RSD value when the MSC method was applied. The SG smoothing-processed data show RSD values with relatively small fluctuation.

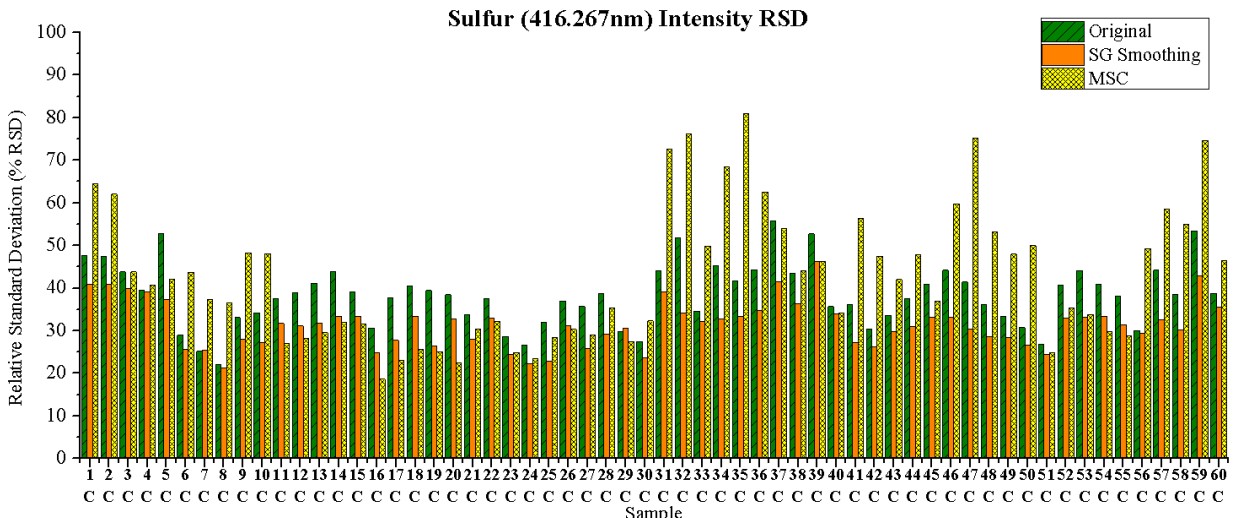

**Figure 3.** RSD result for the major wavelength of sulfur emission line.

### 3.1.2. Partial Least Square Regression (PLSR)

Figure 4a represents the PLSR of original data. The coefficients of determination, $R^2$, for calibration and cross-validation were 0.9265 and 0.8981, respectively. They are larger than or very close to 0.90, which shows a good agreement between calibration and cross-validation. The RMSEC and RMSECV were calculated as 0.0746 and 0.0893, respectively. As shown in Figure 4b, the $R^2$ for calibration, the $R^2$ for cross-validation, the RMSEC, and the RMSECV were improved slightly when the third-order polynomial with five points was used in the SG smoothing method. The $R^2$ of calibration, the $R^2$ of cross-validation, the RMSEC, and the RMSECV were 0.9357, 0.9085, 0.0698, and 0.0847, respectively.

The most widely used third-order polynomial with seven points mode is shown in Figure 4c. More precise values can be seen and compared in Figure 4a,b. The $R^2$s for calibration and cross-validation were 0.9408 and 0.9146, respectively. The RMSEC and RMSECV decreased to 0.0670 and 0.0818, respectively. This means that there is close correlation between the reference data and the estimated data. In Figure 4d, both the $R^2$ for calibration and cross-validation decrease slightly. The $R^2$ of calibration, the $R^2$ of cross-validation, the RMSEC, and the RMSECV were 0.9384, 0.9107, 0.0683, and 0.0837, respectively. This shows that smoothing with many points does not always produce more accurate results. The third-order polynomial with seven points removes only noise but at more points, the information related to the PLSR as well as the noise is lost due to excessive smoothing. Therefore, the estimated *Y* variance dropped below 95% in the nine points mode. The *Y* variance value means how well the variables fitted in the PLSR model and how well they predict new data. Therefore, with regard to the prediction of sulfur content, the SG smoothing mode with third-order polynomial and seven points provided the best result among the above-mentioned four cases.

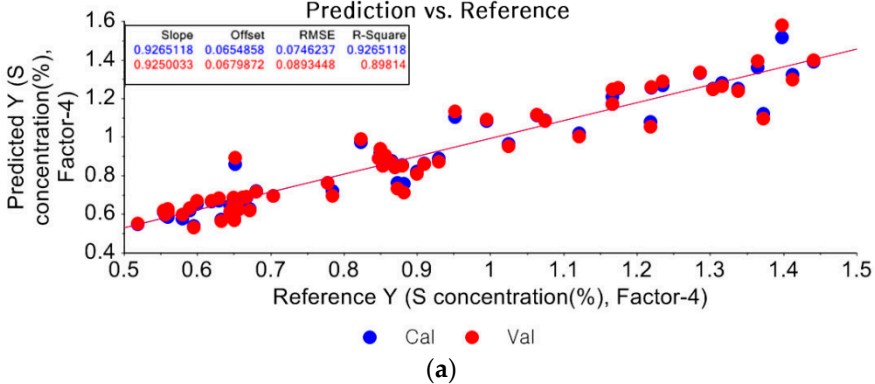

(**a**)

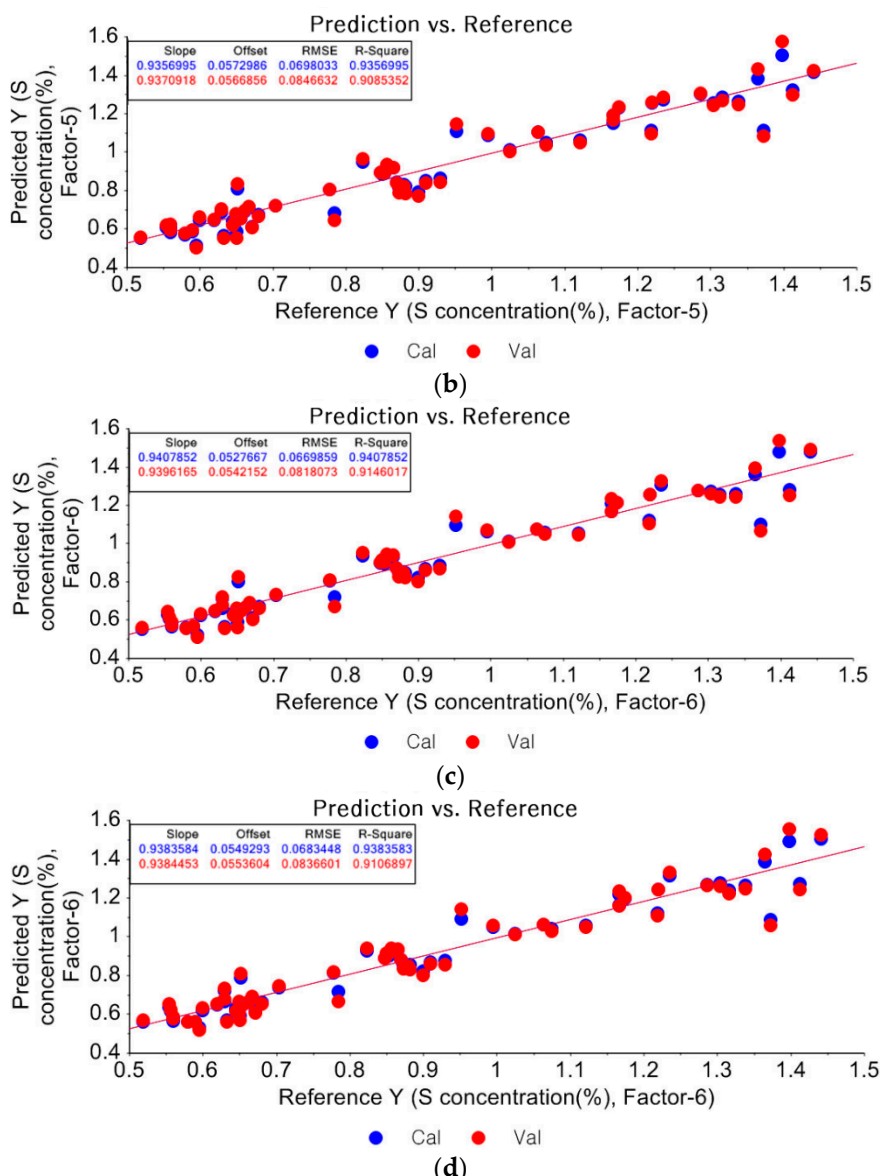

**Figure 4.** PLSR model results (**a**) original data (**b**) the third-order polynomial with five points (**c**) the third-order polynomial with seven points (**d**) the third-order polynomial with nine points.

The cases for the fifth-order polynomial are represented in Figure 5a–d. The $R^2$ of calibration, the $R^2$ of cross-validation, the RMSEC, and the RMSECV for the fifth-order polynomial with seven points were 0.9357, 0.9091, 0.0698, and 0.0844, respectively. In Figure 5b, the $R^2$ of calibration, the $R^2$ of cross-validation, the RMSEC, and the RMSECV for the fifth-order polynomial with nine points were 0.9352, 0.9072, 0.0701, and 0.0853, respectively. In case of the fifth-order polynomial with eleven points, the $R^2$ of calibration, the $R^2$ of cross-validation, the RMSEC, and the RMSECV were 0.9408, 0.9146, 0.067, and 0.0818, respectively. Figure 5d depicts slightly lower values than the data presented in Figure 5c. The $R^2$ of calibration, the $R^2$ of cross-validation, the RMSEC, and the RMSECV for the fifth-order polynomial with thirteen points were 0.9392, 0.9121, 0.0679, and 0.0830, respectively. From the fifth-order mode, the results were improved successively up to the eleven points mode and deteriorated after the thirteen points mode, which caused the $R^2$ value to decrease and the RMSE values to increase. The explained *Y* variance fell below 95% from the thirteen points mode. Since the third-order polynomial with seven points and the fifth-order polynomial with eleven points yielded similar results, the third-order polynomial with seven points was selected as an appropriate data processing mode.

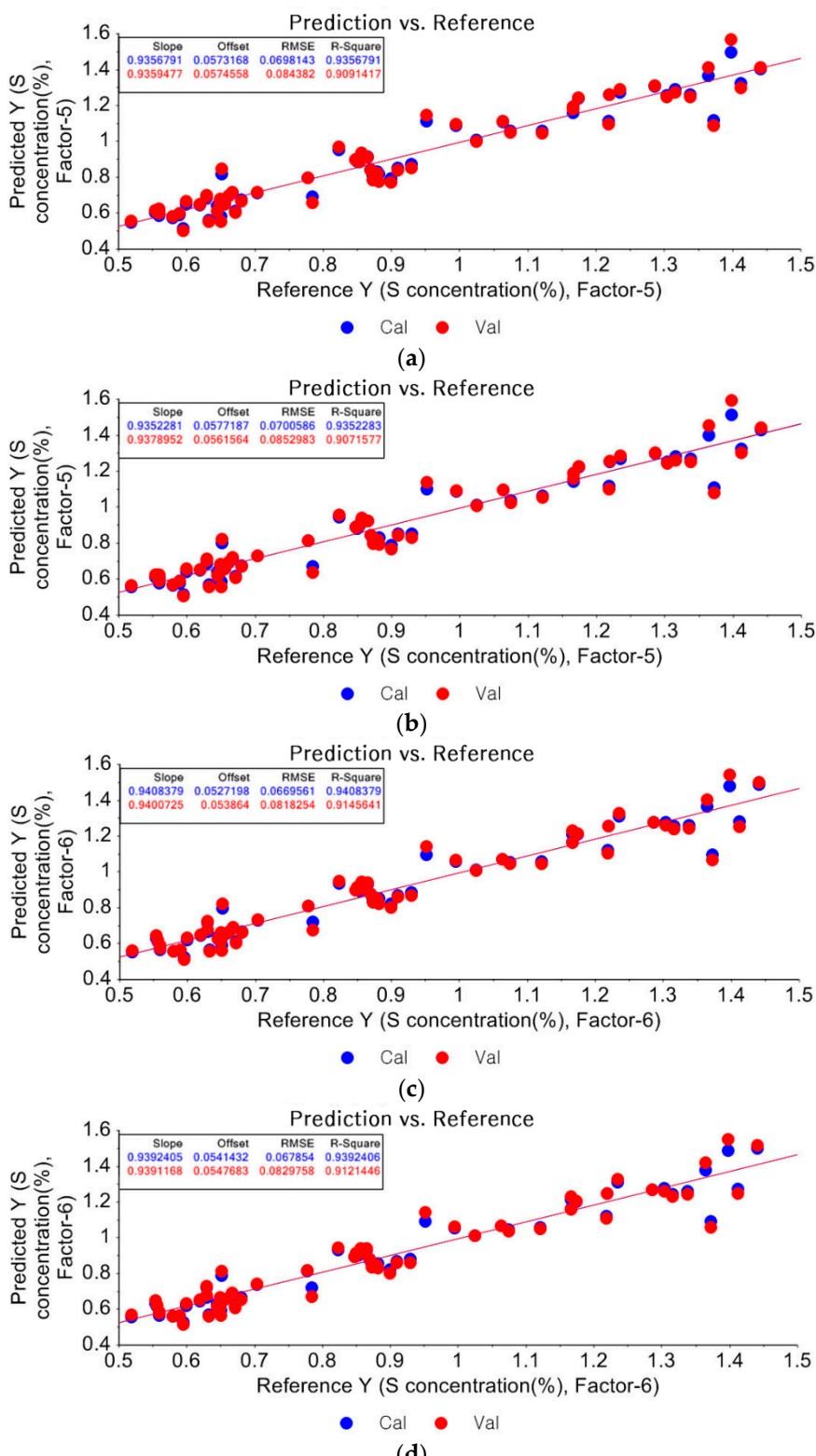

**Figure 5.** PLSR model results (**a**) the fifth-order polynomial with seven points (**b**) the fifth-order polynomial with nine points (**c**) the fifth-order polynomial with eleven points (**d**) the fifth-order polynomial with thirteen points.

Compared with the PLSR results obtained from the original data, PLSR from processed data by the SG smoothing method produced better results in terms of the $R^2$ and the RMSE values. The $R^2$ value of ideal target line is one. This means that the correlation between the predicted value and spectral data is robust as the $R^2$ is close to

unity. The RMSE is reduced by approaching the ideal target line. As the RMSE is smaller, the correlation is closer to the ideal target line. A zero RMSE value means that the predicted value with spectral variables obtained by LIBS exactly matches the measured value by other conventional analysis methods. Therefore, the prediction ability of LIBS can be verified when the RMSE value is close to zero. The $R^2$ values for calibration and cross-validation were improved from 0.9265 to 0.9408 and from 0.8981 to 0.9146, respectively. The RMSEC and RMSECV values were decreased from 0.0746 to 0.0670 and 0.0893 to 0.0818, respectively.

To compare the degree of improvement, the predicted error of sulfur concentration between the PLSR for original data and SG smoothing-processed data is shown in Figures 6 and 7, respectively. The relative error was calculated individually from the reference value and predicted value. As shown in Figure 6, it was found that the maximum relative error value for PLSR of the original data was 24.61%. The error is quite high even though the absolute error was quite low; with a value of 0.1604%. As a result of the sulfur concentration in the coal being so small, even a small change can cause large variation. Therefore, a slight decrease in the error causes a significant improvement in the precision. Compared with the original data, the largest relative error value in processed data with the SG smoothing was 20.44%, as shown in Figure 7. When the data processing method is applied, the largest relative error value could be reduced by 4.174%. Therefore, the accuracy can be improved by processing data in a real-time sulfur concentration measurement system.

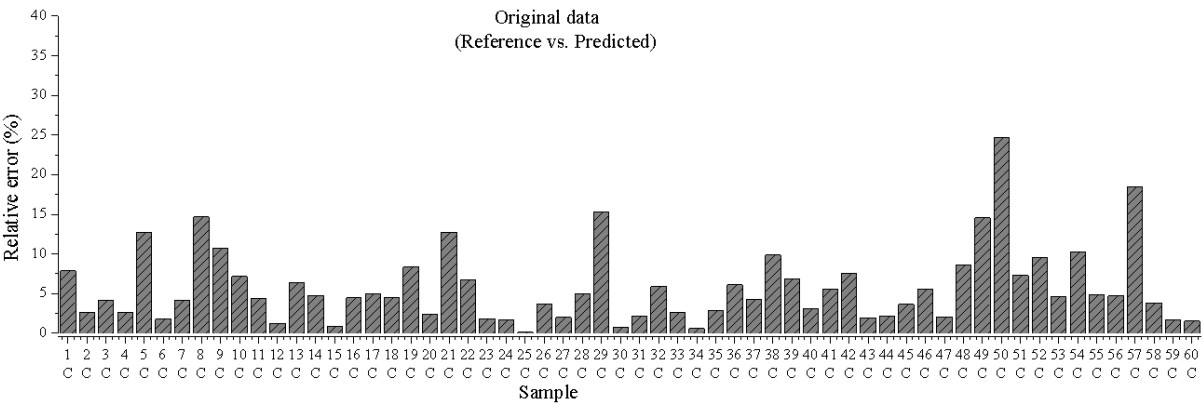

**Figure 6.** Relative error between measured and predicted sulfur concentration for the PLSR of the original data.

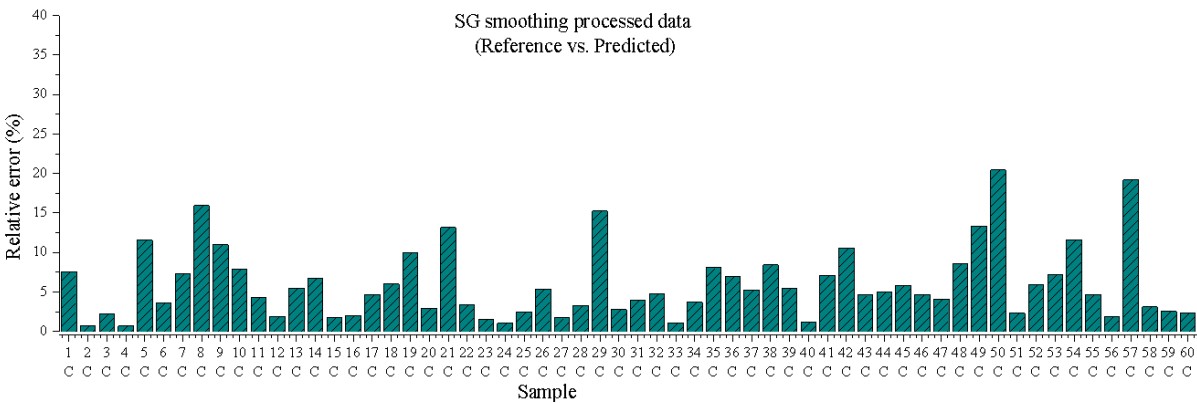

**Figure 7.** Relative error between measured and predicted sulfur concentration for the PLSR of the SG smoothing-processed data.

### 3.1.3. Prediction of Unknown Samples

Prediction plots can be created to evaluate the prediction ability for the unknown samples on the basis of the PLSR results, these are shown in Figure 8. The UO1 and UO2

samples were regarded as an unknown sample even though the sulfur concentration was known to be 0.8800% by air-dried analysis. The UO1 produced a concentration of 0.8878%. This value is calculated as the specifically predicted value from multivariate statistical analysis. There is only 0.0078% difference between the reference value and predicted value in absolute error terms. The relative error value is as small as 0.8864%. In the case of the UO2 sample, the absolute error and relative error values were 0.0998 and 11.341%, respectively. Furthermore, the predictability of unknown blended coal samples was tested. The unknown original sample and another coal sample were mixed to make five blended samples named from UB1 to UB5. Sulfur concentrations in the unknown blended samples are listed in Table 2. The absolute errors and relative errors were calculated by the PLSR model and are indicated in Table 2. Compared to the original unknown sample prediction, the prediction of the blended unknown samples produced similar error values. The produced absolute errors varied from 0.0999 to 0.1492% and the relative errors ranged from 8.8619 to 11.4769%.

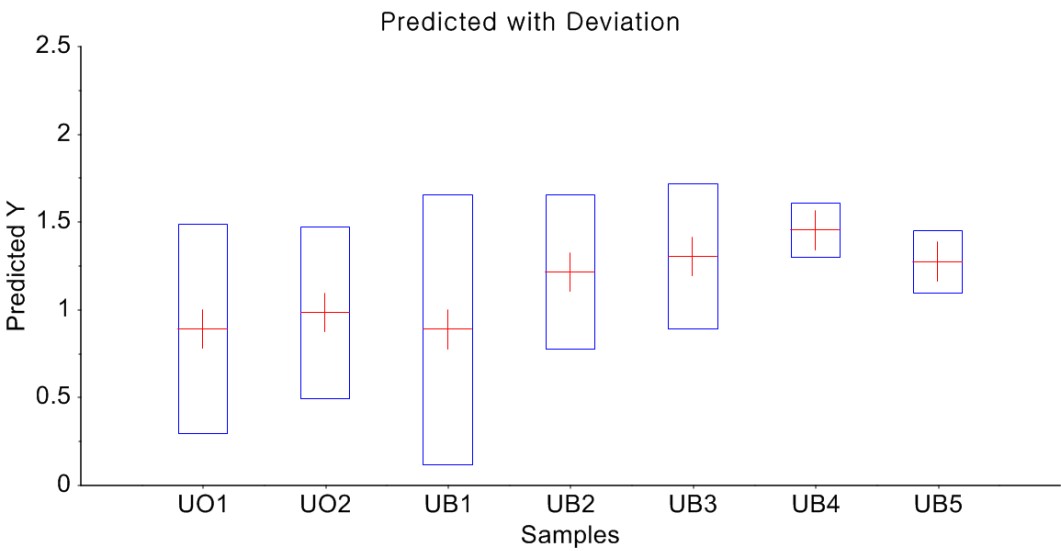

**Figure 8.** Prediction plot of sulfur analysis for the unknown samples.

**Table 2.** Prediction results of sulfur analysis for the unknown samples.

| Sample Number | Measured Value (%) | Predicted Value (%) | Absolute Error (%) | Relative Error (%) |
|---|---|---|---|---|
| UO1 | 0.880 | 0.8878 | 0.0078 | 0.8864 |
| UO2 | 0.880 | 0.9798 | 0.0998 | 11.3409 |
| UB1 | 0.985 | 0.8851 | 0.0999 | 10.1421 |
| UB2 | 1.090 | 1.2115 | 0.1215 | 11.1468 |
| UB3 | 1.195 | 1.3009 | 0.1059 | 8.8619 |
| UB4 | 1.300 | 1.4492 | 0.1492 | 11.4769 |
| UB5 | 1.405 | 1.2706 | 0.1344 | 9.5658 |

### 3.1.4. Root Mean Square Error (RSME) Average

The calculated RMSEC and RMSECV averages for both the original data and the SG smoothing-processed data are indicated in Table 3. These values are used for comparisons between properties of unequal size. The values in this study can be compared with the results of previous studies [4,5]. The RMSEC average for the original data was calculated at 7.6019%. For the RMSECV average value, a slightly higher value was produced, at 9.1453%. In the case of SG smoothing-processed data, improved values were calculated. The RMSEC average in this case was 7.5097% and the RMSECV average was 9.0301%.

**Table 3.** The RMSE average values of the sulfur analysis for the original and SG smoothing processed data.

| Property | Data Process | RMSEC Average (%) | RMSECV Average (%) |
|---|---|---|---|
| Sulfur concentration | Original | 7.6019 | 9.1453 |
| | SG smoothing | 7.5097 | 9.0301 |

*3.2. Calorific Value Analysis*

3.2.1. Partial Least Square Regression (PLSR)

From Dulong's equation, it can be found that carbon, hydrogen, oxygen, and sulfur are the main elements contributing to the higher heating value of coal. Higher heating value, HHV can be theoretically estimated by the following equation [24]:

$$HHV(kcal/kg)=8080C+34460H-4308O+2250S .\qquad(4)$$

The elemental concentration is by weight percent on a dry basis. The PLSR is a linear combination of spectral data and can be used in regression equations. Thus, redundant variables can be removed from data with numerous variables and only the most relevant variants of the spectrum are used in regression analysis. The effect of each variable on the PLSR model can be presented in the form of a regression coefficient plot. The regression coefficient summarizes the relationship between all predictors and the given response. In the LIBS analysis, spectral data is summarized as variables, mainly seven factors. The regression coefficients for these seven factors condense the relationship between the predictors and the response, as a model with seven components approximates it. The information at the wavelength corresponding to the element that greatly influences the property estimation plays an important role. In this study, calorific value will be predicted only using the reference calorific value by PLSR instead of predicting the value for each element and substituting it into Equation (4). Although the use of segments within the full range of the spectrum has been proposed for a better elemental analysis [25], the full range of spectra data was used to preserve informative data when creating the PLSR model. This is because the wavelengths of the elements that significantly affect the calorific value lie in the entire spectral range. The data processing method is essential for a robust calibration model. In the case of sulfur concentration measurements, the third-order polynomial with seven points in the SG smoothing method was used to eliminate noise.

Figure 9a,b represents the PLSR results obtained from the original data for calorific values and the PLSR results of the data processed by SG smoothing, respectively. When compared with the PLSR obtained from the original data, the PLSR of the data processed by the SG smoothing produced better results in terms of the $R^2$ and the RMSE values in the calorific value estimation. A better result can be found in Figure 9b compared to Figure 9a. The $R^2$s for calibration and cross-validation in Figure 9a were 0.9359 and 0.9000, respectively. The RMSEC and RMSECV in Figure 9a were 52.6233 and 66.8441, respectively. In the case of Figure 9b, the $R^2$s for calibration and cross-validation were improved to 0.9472 and 0.9182, respectively. The RMSEC and RMSECV values were decreased to 47.8026 and 60.4917, respectively.

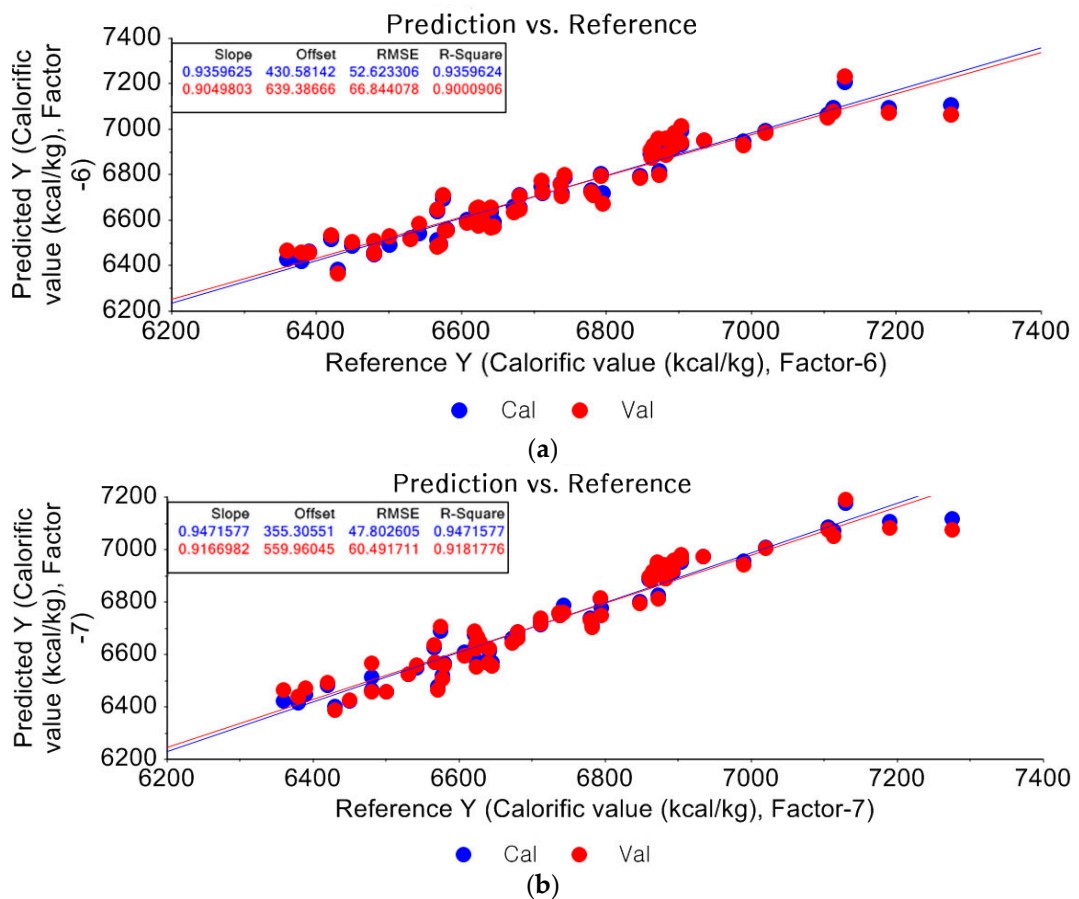

**Figure 9.** PLSR model results (**a**) original data (**b**) the third-order polynomial with seven points SG smoothing method applied.

To compare the degree of improvement, the predicted error of the calorific value between the PLSR for original data and the SG smoothing-processed data are shown in Figures 10 and 11, respectively. The relative error was calculated individually from the reference value and predicted value. There was no dramatic improvement as shown in the case of sulfur concentration estimation, but the error was reduced. The maximum relative error value for PLSR of the original data was 2.3525%. This value is relatively small compared to the error in sulfur concentration analysis. In the case of the processed data with SG smoothing, the largest relative error value was 2.1265%. If the concentration of elements in coal can be accurately predicted in real-time, it can contribute to complete combustion and reduce unburned carbon content by estimating the appropriate amount of combustion air.

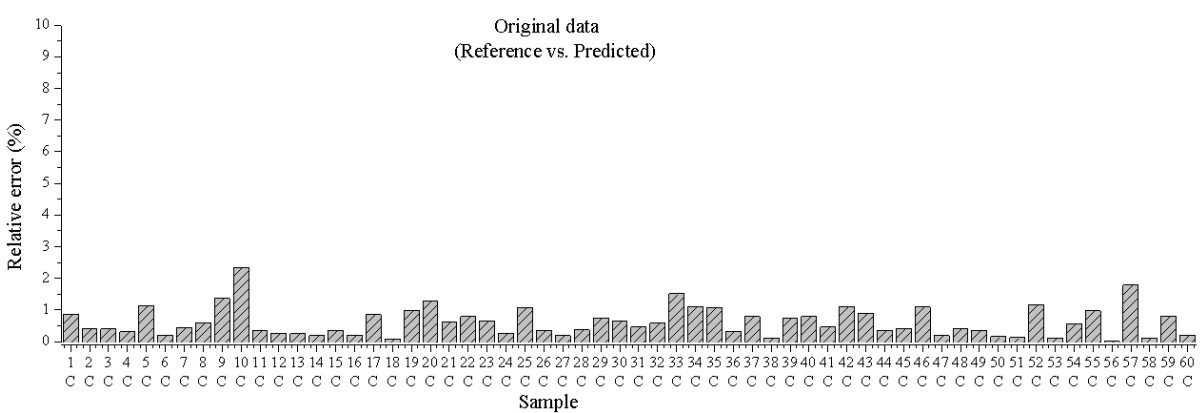

**Figure 10.** Relative error of calorific value between measured and predicted value for the PLSR of original data.

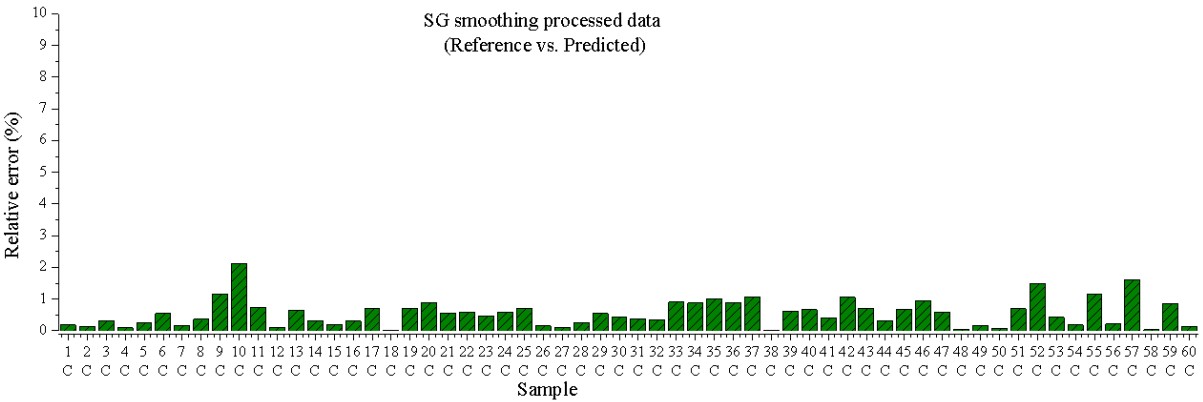

**Figure 11.** Relative error of calorific values between measured and predicted values for the PLSR of SG smoothing-processed data.

3.2.2. Prediction of Unknown Samples

Using the spectral data of an unknown sample, the calorific value can be predicted by the PLSR model. As shown in Figure 12, seven samples were tested. UO1 and UO2 are the unknown samples that were not used in the calibration of the previous PLSR model. The samples named from UB1 to UB5 are the blended unknown samples. The predicted error in calorific values for the unknown samples are listed in Table 4 using the reference calorific value. The calorific values of UO1 and UO2 were known to be 6350 kcal/kg from air-dried basis analysis. As indicated in Table 4, the relative errors range from 0.0663% to 2.7629%. The relative error values of all samples were less than 3%.

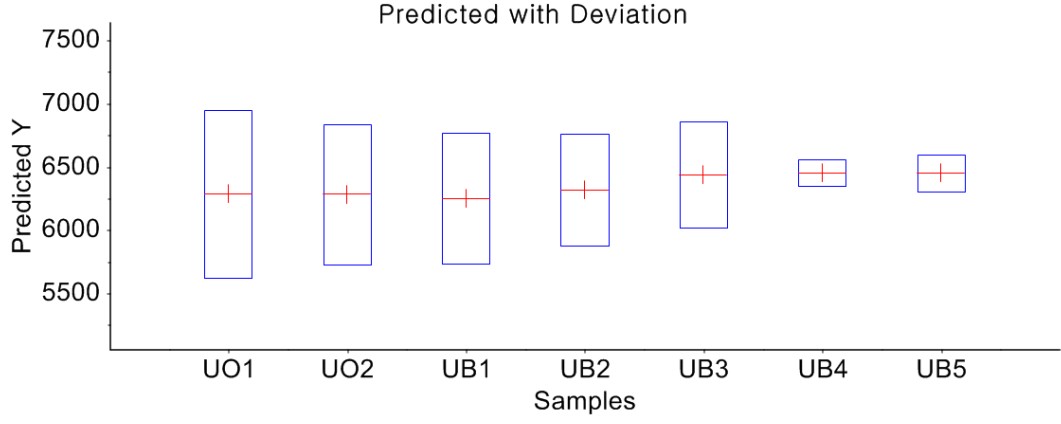

**Figure 12.** Prediction plot of calorific value analysis for the unknown samples.

**Table 4.** Prediction results of calorific values for the unknown samples.

| Sample Number | Measured Value (kcal/kg) | Predicted Value (kcal/kg) | Absolute Error (kcal/kg) | Relative Error (%) |
|---|---|---|---|---|
| UO1 | 6350.000 | 6284.033 | 65.967 | 1.0389 |
| UO2 | 6350.000 | 6279.933 | 70.067 | 1.1034 |
| UB1 | 6376.667 | 6248.880 | 127.787 | 2.0040 |
| UB2 | 6403.333 | 6315.346 | 87.987 | 1.3741 |
| UB3 | 6430.000 | 6434.265 | 4.265 | 0.0663 |
| UB4 | 6456.667 | 6450.932 | 5.735 | 0.0888 |
| UB5 | 6483.333 | 6449.053 | 34.280 | 0.5287 |

### 3.2.3. Root Mean Square Error (RSME) Average

To quantitatively compare the RMSE of the sulfur results, the RMSE averages were calculated by dividing the RMSEC and RMSECV by the average of reference calorific values. The calculated RMSEC and RMSECV averages for both the original data and the SG smoothing-processed data are indicated in Table 5. These values allow comparisons between properties of unequal size. Therefore, the calculated values for calorific values can be compared with the sulfur concentration measurement results. The RMSEC average for the original data and the RMSECV average value for the original data were calculated as 0.7826 and 0.9941%, respectively. This is slightly higher than the RMSEC average value for the original data. In the case of SG smoothing-processed data, more reduced values were calculated. The RMSEC and RMSECV averages were 0.7109 and 0.8997%, respectively. In the case of sulfur analysis, the RMSEC and RMSECV average values of SG smoothing-processed data were calculated as 7.5097 and 9.0301%, respectively. These values are almost ten times higher than the calorific value results. The sulfur content in coal is so small; therefore, the error value tends to increase even with small changes in sulfur concentration. In the case of the calorific value, it can be seen that it is mainly influenced by carbon, referring to Dulong's formula, since the carbon concentration of the coal samples used in this experiment is from 60 to 75%. As a result of the large proportion of carbon in the coal, the prediction errors of calorific value are lower than in the sulfur concentration analysis.

**Table 5.** The RMSE average values of calorific values for the original and SG smoothing-processed data.

| Property | Data Process | RMSEC Average (%) | RMSECV Average (%) |
|---|---|---|---|
| Calorific value | Original | 0.7826 | 0.9941 |
| | SG smoothing | 0.7109 | 0.8997 |

### 4. Conclusions

In this study, LIBS was used to determine the sulfur concentrations and calorific values of blended coals. The PLSR with data processing method helped to reduce the RMSE values and predict the unknown samples. In the RSD calculation, the SG smoothing method showed the lowest value in the largest number of samples and was determined to be an appropriate data processing method. Regarding the relative error, the highest relative error of original data in PLSR for sulfur concentration analysis was 24.61% and this could be reduced to 20.43% as a result of the SG smoothing method. Coal contains a small amount of sulfur compared to other elements; therefore, its concentration is hard to predict accurately and large errors are obtained. Even a slight improvement can be considered to be meaningful in predicting sulfur concentration. In the case of the calorific value analysis, the highest relative error of original data in the PLSR analysis was 2.35% and for the SG smoothing-processed data in the PLSR analysis it was 2.13%. The prediction ability for the unknown samples was evaluated by PLSR analysis. The relative errors in the unknown original sample prediction of sulfur concentration were 3.06% and 4.13%. In addition, the relative errors of the unknown blended sample prediction produced an error value ranging from 5.8 to 14.39%. For the prediction of the calorific value, all relative errors were lower than 3%. When comparing the lowest value found in the reference studies with the largest value of this study, there is only 0.45% difference. The RMSE average values for calorific value measurement were smaller than 1%, which were much smaller than the results of the sulfur concentration measurement. Therefore, this approach will be effective to improve detection ability of the sulfur content in coal for in situ monitoring in a coal-fired power plant. Furthermore, this technique combined with the data processing method is expected to contribute to reducing environmental pollution.

**Author Contributions:** Conceptualization, J.S.C. and S.J.M.; methodology, J.S.C.; software, J.S.C. and C.M.R.; validation, C.M.R.; formal analysis, J.S.C.; investigation, C.M.R.; data curation, C.M.R.; resources, J.H.C.; funding-acquisition, S.J.M.; writing—original draft preparation, J.S.C.; writing—review and editing, J.H.C. and S.J.M.; visualization, C.M.R.; supervision, S.J.M.; All authors have read and agreed to the published version of the manuscript.

**Funding:** This work was partially supported by the Technology Innovation Program (or Industrial Strategic Technology Development Program-Development of technical support platform for welding material and process) (20017251, Development of laser welding automation system for electric coil joining system to manufacture electrical vehicle motors) funded by the Ministry of Trade, Industry & Energy (MOTIE, Korea). This work was partially supported by the National Research Foundation of Korea (NRF) entitled "Development of Coal Analyzing System Using Laser Induced Breakdown Spectroscopy for Clean Coal Power Plant" (No. NRF-2016R1D1A1B03935556). This work was also partially supported by the Basic Science Research Program through the National Research Foundation (NRF) of Korea entitled "NRF-2016R1D1A1B04934910".

**Conflicts of Interest:** The authors declare no conflict of interest. The funders had no role in the design of the study; in the collection, analyses, or interpretation of data; in the writing of the manuscript; or in the decision to publish the results.

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
