# Peer review of "Improving the Analysis of Sulfur Content and Calorific Values of Blended Coals with Data Processing Methods in Laser-Induced Breakdown Spectroscopy"

_applsci, doi:10.3390/app122312410_

Round 1
Reviewer 1 Report
The paper by Moon et al describes a methodology for the estimation of the calorific value of different coal samples using data extracted from LIBS spectra, coupled with a predictive model based on multivariate analysis.
The methods used by the authors are well described and the results are supported by the presented data. These results are presented in a clear way with the effective use of graphs and tables, and the conclusions are well supported by the obtained data.
In the end, I believe that the overall quality of the work presented makes it suitable for publication on the journal, after just a minor revision.
In particular, at PAGE 5 - LINES 184/186, there is a reference to studies on animals or humans and the required ethical authorization. Because this paper does not describe experiments on living beings, I speculate that this text has been erroneously reporte and should be addressed by the authors.
Author Response
Open Review
Comments and Suggestions for Authors
Review 1.
The paper by Moon et al describes a methodology for the estimation of the calorific value of different coal samples using data extracted from LIBS spectra, coupled with a predictive model based on multivariate analysis.
The methods used by the authors are well described and the results are supported by the presented data. These results are presented in a clear way with the effective use of graphs and tables, and the conclusions are well supported by the obtained data.
In the end, I believe that the overall quality of the work presented makes it suitable for publication on the journal, after just a minor revision.
In particular, at PAGE 5 - LINES 184/186, there is a reference to studies on animals or humans and the required ethical authorization. Because this paper does not describe experiments on living beings, I speculate that this text has been erroneously reporte and should be addressed by the authors.
- In the editing process, the sentence was entered incorrectly and was deleted.

Reviewer 2 Report
This work used two well-known data processing methods, i.e. partial least square regression analysis (PLSR) and SG smoothing and MSC method to analyze the data from LIBS. It seems to have better predictions. Some comments can be found as follows.
(1) There are many different data analysis methods. In this work, two specific methods have been used and compared. But if wanting to confirm that the proposed methods are better, other methods should be compared. Honestly, we can find that the reference is mostly old, and no literature is published in the last 5 years.
(2) The structure of this paper should be reorganized. Since the article is focused on data analysis, the basic data analysis method should be presented in the section 2. I don’t understand the data method clearly to be honest.
(3) About the experiment, since you are focused on the sulfur, the related spectral signal should be highlighted in the figure. In Figure 2, I cannot see the signals of sulfur.
(4) The language should be polished. Some typos are “pear” in Line 193, “0.45%p” in Line 461 as examples.
(5) The significant digits should be checked. Too many digits are not so scientific.
Author Response
Review 2.
This work used two well-known data processing methods, i.e. partial least square regression analysis (PLSR) and SG smoothing and MSC method to analyze the data from LIBS. It seems to have better predictions. Some comments can be found as follows.
- There are many different data analysis methods. In this work, two specific methods have been used and compared. But if wanting to confirm that the proposed methods are better, other methods should be compared. Honestly, we can find that the reference is mostly old, and no literature is published in the last 5 years.
- Gazeli, O., Stefas, D., & Couris, S. Sulfur detection in soil by laser induced breakdown spectroscopy assisted by multivariate analysis. Materials 2021, 14(3), 541
-Zhang, W., Zhuo, Z., Lu, P., Tang, J., Tang, H., Lu, J., ... & Wang, Y. LIBS analysis of the ash content, volatile matter, and calorific value in coal by partial least squares regression based on ash classification. Journal of Analytical Atomic Spectrometry 2020, 35(8), 1621-1631.
Above two references were added for recent articles in the manuscript.
(2) The structure of this paper should be reorganized. Since the article is focused on data analysis, the basic data analysis method should be presented in the section 2. I don’t understand the data method clearly to be honest.
- In order to improve the structure, the contents of statistical analysis were added to Chapter 2, and the contents of RSD, PLSR, and RMSE were added.
(3) About the experiment, since you are focused on the sulfur, the related spectral signal should be highlighted in the figure. In Figure 2, I cannot see the signals of sulfur.
- Figure. 2(b) for the sulfur peak was added and explained.
(4) The language should be polished. Some typos are “pear” in Line 193, “0.45%p” in Line 461 as examples.
- Fixed the typo
(5) The significant digits should be checked. Too many digits are not so scientific.
- Since most of the papers using statistical analysis used 4 significant figures, this paper also used 4 significant figures for statistical analysis.
- One unnecessary significant digit was deleted from the calorific value prediction item.

Round 2
Reviewer 2 Report
I am not sure that the authors have detected the spectra of Sulpur in Figure 2. It is not so clear. Could you improve the SNR?
Author Response
I am not sure that the authors have detected the spectra of Sulfur in Figure 2. It is not so clear. Could you improve the SNR?
- Answer: We narrowed down the wavelength range and replotted the Figure 2(b) to distinguish the sulfur peak, clearly. Since we you applied the multivariate method, we did not apply SNR. We added more statement about Figure 2. “Figure 2(a) shows the entire LIBS spectrum of mixed coal sample 4 and 34 as shown in Table 1. The peaks for carbon (247.9 nm), sulfur (416.3 nm), hydrogen (656.3 nm), and oxygen (777.4 nm) are clearly distinguished because they are abundantly included in coal samples. However, as shown in Figure 2(b) the peak (416.267 nm) for sulfur is unclear to be noticed due to the small inclusion in the coal samples. The compositions of S are 0632 and 0.7837 wt% in sample 4 and 34, respectively. The multivariable analysis can be performed based on the maximal intensity of each element.”

Round 3
Reviewer 2 Report
Agree.